# Machine Learning Offers Insights into the Impact of In Vitro Drought Stress on Strawberry Cultivars

**Özhan Şimşek** 

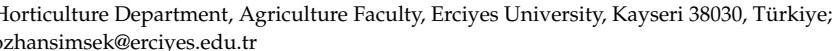

Horticulture Department, Agriculture Faculty, Erciyes University, Kayseri 38030, Türkiye; ozhansimsek@erciyes.edu.tr

**Abstract:** This study aimed to assess the susceptibility of three strawberry cultivars ("Festival", "Fortuna", and "Rubygem") to drought stress induced by varying polyethylene glycol (PEG) concentrations in the culture medium. Plantlets were cultivated on a solid medium supplemented with 1 mg/L BAP, and PEG concentrations (0, 2, 4, and 6 mg/L) were introduced to simulate drought stress. Morphological changes were observed, and morphometric analysis was conducted. Additionally, artificial neural network (ANN) analysis and machine learning approaches were integrated into this study. The results showed significant effects of PEG concentrations on plant height and multiplication coefficients, highlighting genotype-specific responses. This study employed various machine learning models, with random forest consistently demonstrating superior performance. Our findings revealed the random forest model outperformed others with a remarkable global diagnostic accuracy of 91.164%, indicating its superior capability in detecting and predicting water stress effects in strawberries. Specifically, the RF model excelled in predicting root length and the number of roots for "Festival" and "Fortuna" cultivars, demonstrating its reliability across different genetic backgrounds. Meanwhile, for the "Rubygem" cultivar, the multi-layer perceptron (MLP) and Gaussian process (GP) models showed particular strengths in predicting proliferation and plant height, respectively. These findings highlight the potential of ML models, particularly RF, to enhance agricultural breeding and cultivation strategies through accurate phenotypic predictions, suggesting a promising direction for future research to improve these predictions further. This research contributes to understanding strawberry responses to drought stress and emphasizes the potential of machine learning in predicting plant characteristics.

**Keywords:** PEG; BAP; artificial neural network; IBA

## 1. Introduction

Strawberries belong to the category of berry fruits and are characterized by their widespread cultivation across various ecosystems worldwide. They are cultivated in many parts of the world because they can be consumed in a variety of ways, their method of production is appropriate for family businesses, and they provide good income. As they can be produced at different planting times and are grown for both table and industrial production, they demonstrate various characteristics within the species. With these characteristics as goals, breeding programs are conducted in many different world regions [1]. According to Sarıdaş et al. [1], one of the most essential objectives of breeding programs for the cultivation of strawberries is to improve both the yield and quality of the fruits produced by the plants. As the problem of drought spreads across the globe, it is becoming increasingly urgent to identify varieties of strawberries suitable for cultivation in regions with restricted access to irrigation. The evolution of breeding objectives reflects a comprehensive shift towards embracing a holistic approach to agriculture, incorporating not only traditional goals such as increased yield and reduced production costs but also prioritizing product quality, environmental sustainability, and resilience to climate change.

Historically, breeders' primary focus was maximizing yield and minimizing production expenses. However, recent trends have shifted this focus towards enhancing product quality as a critical objective [2,3]. This shift underscores a growing recognition of the interplay between agricultural practices and broader societal needs, including health, nutrition, and environmental stewardship. Integrating product quality into breeding programs introduces a new layer of complexity, necessitating identifying and combining multiple traits within new cultivars [4]. For berry crops, traits that contribute to consumer health and sensory appeal are increasingly vital. These include substances that impart aroma, flavor, and taste, which are influenced by genetic factors and environmental conditions [5]. The challenge extends to understanding the environmental stimuli and signaling mechanisms that promote the accumulation of beneficial bioactive compounds, aligning with consumer demand for health-promoting foods. In response to climate change, breeding programs also incorporate traits that enhance resilience to abiotic stress factors such as drought and high temperatures, alongside resistance to pests and diseases that are expanding their geographic reach [6,7]. This focus on resilience is crucial in the face of projections indicating potential declines in food availability, quality, and nutritional value due to global climate shifts [8,9]. Breeders aim to ensure yield stability and adaptability to diverse environments and cultivation systems by developing cultivars with improved tolerance to biotic and abiotic stresses. The transition of strawberry cultivation from open fields to controlled or protected environments, prompted by regulatory changes such as the Montreal Protocol's ban on certain soil disinfectants, exemplifies the adaptation within agriculture to meet changing production and market demands. This adaptation underscores the need for breeders to offer diverse cultivars suited to various cultivation systems, ensuring that production can meet the growing market demand without compromising quality. As the agricultural sector continues to evolve, the role of breeders becomes increasingly multifaceted. They are tasked with enhancing crops' economic viability and ensuring that agricultural practices contribute positively to human health, nutrition, and environmental sustainability. This comprehensive approach to breeding is essential for addressing the complex challenges of the 21st century, necessitating a balance between productivity, quality, and resilience in the face of changing global conditions [10].

Water scarcity has emerged as a global issue, affecting every region [11]. The 21st century has witnessed a significant rise in the frequency and severity of various natural disasters, including fires and floods. Global warming, which results from the environmental impact that human activities have had on the planet since the industrial revolution, is the most significant contributor to the increase in the frequency of such natural disasters. The evaporation of water is increased by global warming, which ultimately contributes to the severity of droughts. It is reasonable to predict that by the end of the twenty-first century, heat waves will become more frequent and intense [12]. Because of the temperature rise, there is a possibility that some significant changes in atmospheric conditions will occur, including sweltering summers and periods of drought [13]. These climate changes affect trophic interactions, species distribution, and ecosystem function. It is anticipated that future stresses such as drought and elevated temperatures will worsen due to the rapid growth of the human population. Temperatures that are too high and a lack of available water are two significant interrelated stresses that affect growth and productivity and ultimately affect the safety of food supplies [14]. In this context, it is vital to determine the molecular physiological mechanisms and signaling pathways responsible for increased drought tolerance to be better prepared for future drought stress effects.

Strain caused by drought is rapidly becoming a more widespread issue worldwide. The quantity and quality of produce obtained from aquaculture are affected when there is a high concentration of agricultural land and greenhouse cultivation in dry and semi-dry regions worldwide [15]. The strawberry is a species that is overly sensitive to the effects of drought stress; under these conditions, it quickly loses its turgor, and the yield and plant growth both suffer due to the restriction of photosynthesis. Strawberries can tolerate various levels of water stress depending on factors such as the plant's development stage,

duration of the stress, growing system, growing environment, and variety. In one study, strawberries grown in a greenhouse were compared with those grown outside. It was found that strawberries grown without soil are more susceptible to water stress than soil-grown strawberries [16]. The most significant effect of water stress on strawberry varieties is a reduction in fruit size and yield; however, the exact nature and magnitude of this reduction vary from variety to variety [16].

Genotypic differences in drought tolerance have been analyzed in various crop species, including strawberries [17–25]. However, there is still a lack of information regarding the morphophysiological changes in various in vitro cultured strawberry cultivars when subjected to limited water availability [20]. Simulating osmotic stress in vitro using osmotic agents like high-molecular-weight polyethylene glycol (PEG) is a widely adopted method for researching plant responses to water stress. PEG is advantageous because it is highly soluble, does not penetrate cells, and creates a negative osmotic potential without inducing toxicity. This approach allows for a controlled examination of water stress mechanisms, offering insights into plant resilience and adaptation strategies under drought conditions, thus facilitating targeted improvements in crop management and breeding programs for enhanced drought tolerance [26].

In computer science, machine learning is commonly recognized as an example of artificial intelligence. Using datasets for training enables computers to absorb knowledge [27]. Machine learning (ML) entails building strong mathematical models using datasets containing various independent variables or components and dependent variables or responses [28]. Machine learning algorithms have the potential to be effective and predictive decision-making tools for in vitro plant micropropagation processes because of their ability to forecast and define complex processes involving numerous components [29,30]. However, compared with their widespread application in other scientific fields, the application of ML techniques in the context of plant and agricultural sciences is somewhat limited [31]. Artificial neural networks (ANNs) are a class of nonlinear computing methods used for various tasks such as clustering data, generating predictions, and classifying complex systems [32].

Recently, diverse machine learning models have proven effective in accurately forecasting and refining plant tissue culture procedures. These models have been applied in various investigations, including in vitro mutagenesis, micropropagation, regeneration studies, plant system biology, in vitro organogenesis, stress physiology, and salt stress [28–42]. Only a few studies have used machine learning models to examine drought stress responses. In a representative study, Jafari and Shahsavar [32] used artificial neural networks to simulate and forecast the effects of melatonin on the morphological responses of citrus plants under drought stress. Recent studies demonstrate innovative approaches to drought stress detection and analysis in agriculture. Gupta et al. [33] developed an automation model for water stress detection in wheat using pre-processing and canopy segmentation methods, with the random forest algorithm achieving high accuracy. Das Choudhury et al. [34] introduced HyperStressPropagateNet, a deep neural network for analyzing drought stress propagation in plants through hyperspectral imagery, showing a strong correlation with soil water content. Tahmasebi et al. [35] applied a meta-analysis and machine learning to identify drought-responsive genes in *Populus*, revealing significant transcriptional variations and potential markers for breeding programs. These contributions highlight the role of advanced computational methods in enhancing our understanding of plant responses to drought stress.

The primary objective of this study was to evaluate the susceptibility to drought of three different strawberry cultivars grown in vitro. The introduction of varying concentrations of PEG into the culture medium caused morphological changes in the plants, which were then subjected to morphometric analysis for quantification. This study aimed to integrate artificial neural network (ANN) analysis and machine learning approaches to enhance the research scope. Computational techniques were used to model and predict the effects of various culture media components on micropropagation quality. In this study, we

used three machine learning (ML) algorithms—Gaussian process (GP), random forest (RF), and support vector machine (SVM).

## 2. Materials and Methods

### 2.1. Plant Material and Surface Sterilization

In this study, the strawberry cultivars "Rubygem", "Fortuna", and "Festival" served as the source of the plant material. Shoot tips were given a 10 min rinse under running tap water. The tips of the shoots were then soaked for 15 min in a solution consisting of 15 percent sodium hypochlorite and 1–2 drops of Tween 20. Subsequently, the sterilant materials were removed by three separate washes in distilled sterile water.

### 2.2. Establishment of In Vitro Drought Stress

The effects of drought were investigated using shoot tips collected from different strawberry cultivars. MS media containing 0%, 2%, 4%, and 6% PEG 6000 were prepared to achieve this goal, and the plants were then transferred to these media. Plant micropropagation was facilitated by the addition of 1 mg/L BAP to the MS nutrient medium [43]. The plants were examined weekly for four weeks, and three subcultures were established. The duration of the subculture was four weeks. The plants' responses to drought on their terms were investigated in each subculture period. To accomplish this goal, measurements of the shoot length (in centimeters) and the proliferation rate (number of proliferated shoots from the starting explant) were taken during every subculture period that occurred during the culture period.

After completing the third subculture and rooting experiment, strawberry plants were transplanted into a rooting medium. In the rooting experiment, preparations were made to use media containing PEG 6000 to maintain drought stress. Each media had 30 g/L of sucrose, 5.7–5.8 pH, 4.4 g/L of MS [43], and 1 mg/L of IBA (indole butyric acid) added to it. The pH of the medium was adjusted to be between 5.7 and 5.8. The medium was supplemented with PEG at various concentrations, ranging from 0% to 6%, to maintain drought stress. After adding PEG, 8 g/L agar was added. The medium was heated to 121 °C and subjected to 1.05 atmospheres of pressure during sterilization. After the setting process, the plants were transferred to the rooting medium. Plants were grown in a chamber room at 25 ± 2 °C with 16 h of light and 8 h of darkness. After a six-week cultivation period for in vitro rooting, the ability of plants to form new roots was evaluated. Several parameters, including root number, average root length (cm), rooting rate (percent), and plant height (cm), were investigated.

### 2.3. Statistical Analysis

Micropropagation and rooting experiments of "Rubygem", "Fortuna", and "Festival" strawberry cultivars under in vitro drought stress conditions were established in a factorial order random plot trial design. There were three replicates for each cultivar and ten plants in each replicate. An analysis of variance (ANOVA) was conducted with the data obtained. Following the analysis of variance, the software programs in which the changes had a significant impact were subjected to the LSD test. Subsequently, the percentage change relative to the control (PEG 0) was determined for each parameter across the PEG treatments, allowing for a comparative analysis of drought stress impact. The JMP program was used to carry out statistical analyses.

### 2.4. Modeling Procedure

In this study, we used three machine learning (ML) algorithms—Gaussian process (GP), random forest (RF), and support vector machine (SVM)—as well as two well-known artificial neural network (ANN)-based multilayer perceptrons (MLPs) to model and predict the micropropagation and rooting efficiency of various strawberry genotypes through PEG. We divided the dataset into training and testing subsets using a 10-fold cross-validation method to thoroughly evaluate the predictive performances of MLP and ML models.

Three different genotypes were included in the input variables, with an additional 2 mm/L of PEG functioning as an input variable. Conversely, plant height, root number, root length, and multiplication rate comprised the target (output) variables. R-programming was used to implement coding with the help of the Caret and Kernlab packages. Several metrics were used to assess and compare the accuracy and precision of MLP and ML models. These metrics included the coefficient of determination ($R^2$), which shows the degree of relationship between the model and dependent variable; root mean square error (*RMSE*), which shows how closely the regression line matches the observed data points; and mean absolute error (*MAE*), which calculates the average error between the predicted and observed values.

### 2.4.1. Support Vector Machines

Support vector machines (SVMs), developed by Vapnik [44], refer to a family of artificial intelligence models that include supervised and unsupervised learning methods. These models are well-suited for regression analysis, clustering, and classification [45]. In this section, we focus on their application in regression tasks. Support vector regression (SVR), introduced here, utilizes the SVM framework for regression purposes. Unlike traditional artificial intelligence algorithms, SVM techniques, particularly SVR, demonstrate efficiency even with relatively small datasets, making them well-suited for regression analysis. This efficiency is crucial, especially when dealing with limited training data. Furthermore, SVMs, and SVR in particular, address common issues associated with other algorithms, such as overfitting, low convergence rates, and entrapment in local minima. Equation (1) declares the SVM algorithm in the context of regression, aiding in determining the optimal separator plane for regression tasks:

$$f(x) = w\varphi(x) + b \tag{1}$$

### 2.4.2. Random Forest

The RF ensemble learning technique, which is essentially an ensemble of unpruned trees, was invented by Breiman [46]. Regression and classification tasks have both shown success with RF, which is well known for its superior efficiency and ease of design. The prominent characteristics of the RF model supported by previous research include its ability to avoid overfitting, proficient handling of noise, and efficient management of many features [47].

Two random sources were injected into each tree inside the RF model for classification accuracy during construction to minimize the correlation between distinct trees while maintaining their unique strengths. Each tree was first trained by randomly selecting a replacement (bootstrap replica) of the training set. The algorithm then considered a small variable subset randomly chosen from the complete variable set to find the optimal split at each node. Additionally, every tree was completely mature, producing low bias and significant variance in tree outputs [45,47]. Regression models were solved using the mean squared error (MSE) metric, which was used to determine the distance between nodes and the best branching choices within the forest. Equation (2) clarifies the fundamental idea:

$$y = \sum_{i=1}^{n} (a_i - a_i^*)k(x, x_i) + b \tag{2}$$

### 2.4.3. Gaussian Process

To better understand the spread of random variables, the Gaussian process (GP) model for supervised learning expands the Gaussian probability distribution. This makes the GP model more appropriate for solving the classification and regression issues. By calculating the likelihood that the input samples fall into particular classes, it performs the role of a nonparametric regressor for datasets. The ability of this model to work well with tiny datasets is one of its main advantages: it offers consistency, accuracy, and ease of

computing [48]. The derivation method for each input (*x*) and its matching output (*y*) is outlined in Equation (3):

$$y_i = f(x_i) + \varepsilon \tag{3}$$

### 2.4.4. Multilayer Perceptron

One well-known artificial neural network (ANN) is the multilayer perceptron (MLP). It is arranged in layers with an input layer, an output layer, and one or more hidden layers. Using a supervised training technique, the MLP was trained using input and output variables from the training set. The training cycle is repeated until the reduction in Equation (4) is accomplished:

$$E = \frac{1}{K} \sum_{k=1}^{K} (y_{sk} - \hat{y}_k)^2 \tag{4}$$

*K* = number of samples;
*y* = observed value of data point *k*.

To calculate the predicted value $\hat{y}$ in the multilayer perceptron (MLP), which has a hidden layer with p neurons and k output variables, the following equation was applied:

$$\hat{y} = f\left[ \sum_{j=1}^{p} w_{ji}.g(\sum_{i=1}^{k} w_{ji}x_i + w_{j0}) + w_o \right] \tag{5}$$

$x_i$ represents the ith output variable, $w_j$ corresponds to the weighted input data entering the jth hidden neuron, *f* is the activation function applied to the output neuron, $w_{ji}$ signifies the weight associated with the direct connection from input neuron i to hidden neuron *j*, $w_{j0}$ represents the bias specific to the *j*th neuron, $w_0$ represents the bias linked to $\hat{y}$ the output neuron, and g is the activation function utilized for the hidden neuron.

## 3. Results

Three different genotypes of strawberries ("Festival", "Fortuna", and "Rubygem") were cultured on their shoot tips on a solid medium supplemented with 1 mg/L BAP. Different concentrations of polyethylene glycol (PEG) (0, 2, 4, and 6 mg/L) were added to these media to simulate drought stress. The resultant plantlets were meticulously measured for each plant's height in centimeters after cultivation. Table 1 presents the tabulated results, which summarize the observed differences in plant heights.

**Table 1.** Plant height (cm).

| | PEG-free | PEG 2 mg/L | PEG 4 mg/L | PEG 6 mg/L | Cultivar Average |
|---|---|---|---|---|---|
| "Festival" | 4.26 | 3.60 | 3.06 | 2.35 | 3.32 AB |
| "Fortuna" | 4.70 | 3.72 | 3.23 | 2.46 | 3.53 A |
| "Rubygem" | 4.03 | 3.37 | 2.46 | 2.39 | 3.18 B |
| PEG average | 4.33 A | 3.57 B | 3.09 C | 2.40 D | |
| Variation in percentage | 0.00 | −17.10 | −28.58 | −44.58 | |

$LSD_{Cultivar}$: 0.382 *, $LSD_{PEG}$:0.285 ***, $LSD_{Cultivar*PEG}$: N.S. *: $p < 0.05$, ***: $p < 0.001$, capital letters indicate statistical differences between strawberry varieties and different PEG concentrations, as determined by the LSD analysis.

According to Table 1, it is noteworthy that the cultivar "Fortuna" reached its maximum plant height (4.70 cm) when grown in a PEG-free (0 mg/L) medium. In contrast, the cultivar "Festival" grown in a growth medium containing 6 mg/L PEG showed the shortest plant height of all the genotypes under investigation, measuring 2.35 cm. These findings highlight the critical effects of PEG concentrations on the growth traits of the strawberry cultivars under investigation and provide insight into the complex reactions of various genotypes to changing osmotic conditions. The analysis indicated a progressive decline in plant height

with increasing PEG concentrations, showcasing a decrease of 17.10% at 2 mg/L PEG, deepening to 28.58% at 4 mg/L PEG, and reaching a substantial reduction of 44.58% at 6 mg/L PEG. This trend underscores the adverse effects of escalating drought stress on the vertical growth of strawberry plants.

The cultivars "Fortuna" and "Festival" showed the best proliferation coefficients, with rates of 5.20 when grown in a growth medium free of PEG. In comparison, the "Fortuna" and "Rubygem" cultivars, when cultivated in a growth medium containing 6 mg/L PEG, had the lowest multiplication coefficient, 2.80. These findings highlight how PEG concentrations significantly affect the dynamics of strawberry cultivar multiplication. The multiplication rate experienced significant reductions under drought conditions, decreasing by 17.56% at 2 mg/L PEG, further declining to 34.15% at 4 mg/L PEG, and exhibiting the most pronounced drop of 44.88% at 6 mg/L PEG. These findings highlight the detrimental impact of drought stress on the plant's reproductive capacity and overall vitality (Table 2).

**Table 2.** Proliferation rate of strawberry cultivars.

| | PEG-free | PEG 2 mg/L | PEG 4 mg/L | PEG 6 mg/L | Cultivar Average |
|---|---|---|---|---|---|
| "Festival" | 5.20 | 4.25 | 3.35 | 2.85 | 3.91 |
| "Fortuna" | 5.20 | 4.20 | 3.60 | 2.80 | 3.95 |
| "Rubygem" | 4.90 | 4.20 | 3.20 | 2.80 | 3.77 |
| PEG average | 5.10 A | 4.22 B | 3.38 C | 2.82 D | |
| Variation in percentage | 0.00 | −17.56 | −34.15 | −44.88 | |

$LSD_{Cultivar}$: N.S, $LSD_{PEG}$: 0.479 ***, $LSD_{Cultivar*PEG}$: N.S., ***: $p < 0.001$, capital letters indicate statistical differences between strawberry varieties and different PEG concentrations, as determined by the LSD analysis.

The root numbers at various PEG concentrations of PEG in Table 3 indicate unique patterns among the strawberry varieties under investigation. "Fortuna" and "Rubygem" had the most significant root number 5.20) in the 6 mg/L PEG, followed by "Festival" (5.05) and "Fortuna" (3.70).

**Table 3.** Number of roots.

| | PEG-free | PEG 2 mg/L | PEG 4 mg/L | PEG 6 mg/L | Cultivar Average |
|---|---|---|---|---|---|
| "Festival" | 4.50 | 4.75 | 3.80 | 5.05 | 4.52 |
| "Fortuna" | 4.80 | 4.70 | 3.90 | 5.20 | 4.65 |
| "Rubygem" | 4.50 | 4.70 | 3.90 | 5.20 | 4.57 |
| PEG average | 4.60 A | 4.71 A | 3.86 B | 5.15 A | |
| Variation in percentage | 0.00 | 3.28 | −15.85 | 12.02 | |

$LSD_{Cultivar}$: N.S, $LSD_{PEG}$:0.58 ***, $LSD_{Cultivar*PEG}$: N.S., ***: $p < 0.001$, capital letters indicate statistical differences between strawberry varieties and different PEG concentrations, as determined by the LSD analysis.

However, all cultivars showed a discernible increase in the root count at the maximum PEG concentration (6 mg/L). At 4.65, "Fortuna" showed the most significant average root numbers, closely followed by "Rubygem" at 4.57, while "Festival" was at 4.52. The response of root production to drought stress was more variable. At 2 mg/L PEG, a slight increase of 3.28% was observed, suggesting a potential initial compensatory root proliferation response to mild drought. However, at 4 mg/L PEG, a decrease of 15.85% was noted, with a subsequent increase of 12.02% at 6 mg/L PEG, indicating complex adaptive responses in root system architecture under varying levels of water availability.

According to Table 4, root rates at various concentrations of polyethylene glycol (PEG), considering transformation rates within parentheses, offer valuable information on the response characteristics of strawberry cultivars under investigation.

**Table 4.** Rooting rate of strawberry cultivars.

| | PEG-free | PEG 2 mg/L | PEG 4 mg/L | PEG 6 mg/L | Cultivar Average |
|---|---|---|---|---|---|
| "Festival" | 80 (72.00) | 65 (58.50) | 90 (81.00) | 85 (76.50) | 80 (72.00) |
| "Fortuna" | 80 (72.00) | 60 (54.00) | 90 (81.00) | 70 (63.00) | 75 (67.00) |
| "Rubygem" | 80 (72.00) | 70 (63.00) | 70 (63.00) | 80 (72.00) | 75 (67.00) |
| PEG average | 80 (72.00) | 65 (58.50) | 83 (75.00) | 78 (70.50) | |
| Variation in percentage | 0.00 | −18.75 | 6.25 | 0.00 | |

LSD$_{Cultivar}$: N.S, LSD$_{PEG}$: N.S, LSD$_{Cultivar*PEG}$: N.S.

Regarding transformation rates inside parentheses, the "Festival" cultivar, at 90 (81.00), had the highest root rate, noted in the presence of 4 mg/L PEG. On the other hand, at 2 mg/L PEG, the "Fortuna" cultivar showed the lowest root rate, measuring 60 (54.00).

To predict in vitro plant characteristics, such as plant height, proliferation, root number, and root length, four different machine learning (ML) and artificial neural network (ANN) models were used: multilayer perceptron (MLP), support vector machine (SVM), Gaussian process (GP), and random forest (RF). The root mean square error (*RMSE*), coefficient of determination ($R^2$), and mean absolute error (*MAE*) metrics were used to evaluate the validity of the model Equations (6)–(8). The $R^2$ values range from 0 to 1, where 1 denotes an ideal prediction and 0 denotes no capacity for explanation. Model precision is represented by the *RMSE* values, which generally range from zero to positive infinity. Lower values indicate better performance. Similarly, *MAE* represents the predicted accuracy and ranges from zero to positive infinity, where lower values denote higher accuracy.

$$R^2 = 1 - \frac{\sum_{i=1}^{n}(Y_i - \hat{Y}_i)^2}{\sum_{i=1}^{n}(Y_i - \tilde{Y})^2} \tag{6}$$

$$RMSE = \sqrt{\frac{(\sum_{i=1}^{n}(Y_i - \hat{Y}_i)^2)}{n}} \tag{7}$$

$$MAE = \frac{1}{n}\sum_{i=1}^{n}\left|Y_i - \hat{Y}_i\right| \tag{8}$$

A higher agreement between the expected and actual values is shown by an $R^2$ value closer to 1 and *RMSE* and *MAE* values closer to 0.

Table 5 presents the models' $R^2$, *MAE*, and *RMSE* values for plant height, proliferation, number of roots, and root length for all strawberry cultivars. The results reveal the relatively close $R^2$ values of all models for plant height, except the MLP model; these values were 0.66 (RF, GP) and 0.64 (SVM), while MLP showed the lowest $R^2$ at 0.52. *RMSE* and *MAE* values varied with the model and exhibited the order RF (0.55) < GP (0.56) < SVM (0.57) < MLP (0.72) for plant height. *MAE* also showed the same order as *RMSE*, with RF having the lowest *MAE* and MLP having the highest. The $R^2$, *MAE*, and *RMSE* values for proliferation in all tested and recorded models were in the order RF (0.78; 0.75; 0.57) > GP (0.60; 0.61; 0.75) > SVM (0.59; 0.59; 0.76) > MLP (0.55; 0.71; 0.91). It can be deduced from these results that the RF model was the best among the tested models for proliferation owing to its highest $R^2$ (0.78) and relatively lowest *RMSE* (0.57) scores. The $R^2$, *MAE*, and *RMSE* values for the number of roots in all tested and recorded models were in the order RF (0.84; 0.64; 0.86) > SVM (0.72; 0.90; 1.16) > MLP (0.68; 0.95; 1.27) > GP (0.61; 0.96; 1.24). The RF model performed the best in terms of predicting the number of roots. The same thing happened with $R^2$, *MAE*, and *RMSE* values for root length; all tested and recorded models performed in the order RF (0.89; 0.46; 0.63) > MLP (0.84; 0.59; 0.85) > GP (0.73; 0.82; 1.07) > SVM (0.70; 0.77; 107). The RF model outperformed the other models in terms of root length.

The overall performance of the models based on all variables (plant height, proliferation, number of roots, and root length) can be ranked in the order RF > GP > SVM > MLP.

**Table 5.** Validation of machine learning model algorithms for three strawberry cultivars.

|  |  | Plant Height | Proliferation | Number of Roots | Root Length |
|---|---|---|---|---|---|
| **MLP** | *RMSE* | 0.72 | 0.91 | 1.27 | 0.85 |
|  | $R^2$ | 0.52 | 0.55 | 0.68 | 0.84 |
|  | *MAE* | 0.55 | 0.71 | 0.95 | 0.59 |
| **SVM** | *RMSE* | 0.57 | 0.76 | 1.16 | 1.07 |
|  | $R^2$ | 0.64 | 0.59 | 0.72 | 0.70 |
|  | *MAE* | 0.44 | 0.59 | 0.90 | 0.77 |
| **RF** | *RMSE* | 0.55 | 0.57 | 0.86 | 0.63 |
|  | $R^2$ | 0.66 | 0.78 | 0.84 | 0.89 |
|  | *MAE* | 0.41 | 0.44 | 0.64 | 0.46 |
| **GP** | *RMSE* | 0.56 | 0.75 | 1.24 | 1.07 |
|  | $R^2$ | 0.66 | 0.60 | 0.61 | 0.73 |
|  | *MAE* | 0.44 | 0.61 | 0.96 | 0.82 |

We also conducted ML prediction per each cultivar. Table 6 presents the models' *RMSE*, $R^2$, and *MAE* values for plant height, proliferation, number of roots, and root length of the "Rubygem" cultivar. $R^2$ values for plant height scored 0.87 for GP, 0.85 for MLP, and 0.78 for RF, while SVM showed the lowest $R^2$ at 0.70. The *RMSE* and *MAE* values varied by model and exhibited the order GP (0.11) < RF (0.13) = SVM (0.13) = MLP (0.13) for plant height. *MAE* also showed the same order as *RMSE*, with GP having the lowest *MAE*. In terms of *RMSE*, $R^2$, and *MAE* values for proliferation, all tested and recorded models performed in the order MLP (0.16; 0.90; 0.14) > GP (0.16; 0.82; 0.12) > RF (0.16; 0.81; 0.13) > SVM (0.15; 0.66; 0.13). It can be deduced from these results that the MLP model was the best among the tested models in terms of predicting proliferation owing to its highest $R^2$ (0.90). The *RMSE*, $R^2$, and *MAE* values for the number of roots in all tested and recorded models were in the order MLP (0.26; 0.57; 0.23) > SVM (0.28; 0.57; 0.24) > GP (0.26; 0.54; 0.23) > RF (0.26; 0.51; 0.23). For the number of roots, MLP also showed the highest $R^2$. The same thing happened with *RMSE*, $R^2$, and *MAE* values for root length; all tested and recorded models performed in the order RF (0.89; 0.46; 0.63) > MLP (0.84; 0.59; 0.85) > GP (0.73; 0.82; 1.07) > SVM (0.70; 0.77; 107). The RF model outperformed the other models in terms of root length. The overall performance of the models based on all variables (plant height, proliferation, number of roots, and root length) for the "Rubygem" cultivar can be ranked in the order MLP > GP > RF > SVM.

**Table 6.** Validation of machine learning model algorithms for "Rubygem" cultivar.

|  |  | Plant Height | Proliferation | Number of Roots | Root Length |
|---|---|---|---|---|---|
| **MLP** | *RMSE* | 0.13 | 0.16 | 0.26 | 0.21 |
|  | $R^2$ | 0.85 | 0.90 | 0.57 | 0.45 |
|  | *MAE* | 0.11 | 0.14 | 0.23 | 0.18 |
| **SVM** | *RMSE* | 0.13 | 0.15 | 0.28 | 1.07 |
|  | $R^2$ | 0.70 | 0.66 | 0.57 | 0.44 |
|  | *MAE* | 0.11 | 0.13 | 0.24 | 0.77 |
| **RF** | *RMSE* | 0.13 | 0.16 | 0.26 | 0.23 |
|  | $R^2$ | 0.78 | 0.81 | 0.51 | 0.58 |
|  | *MAE* | 0.11 | 0.13 | 0.23 | 0.21 |
| **GP** | *RMSE* | 0.13 | 0.16 | 0.26 | 0.22 |
|  | $R^2$ | 0.87 | 0.82 | 0.54 | 0.34 |
|  | *MAE* | 0.10 | 0.12 | 0.23 | 0.18 |

MLP: multilayer perceptron; SVM: support vector machine; RF: random forest; GP: Gaussian process; $R^2$: coefficient of determination; *MAE*: mean absolute error; *RMSE*: root mean square error.

The actual and predicted values of "Rubygem" are compared in Figure 1. The samples are shown on the horizontal axis, whereas the model's predicted findings are on the vertical axis.

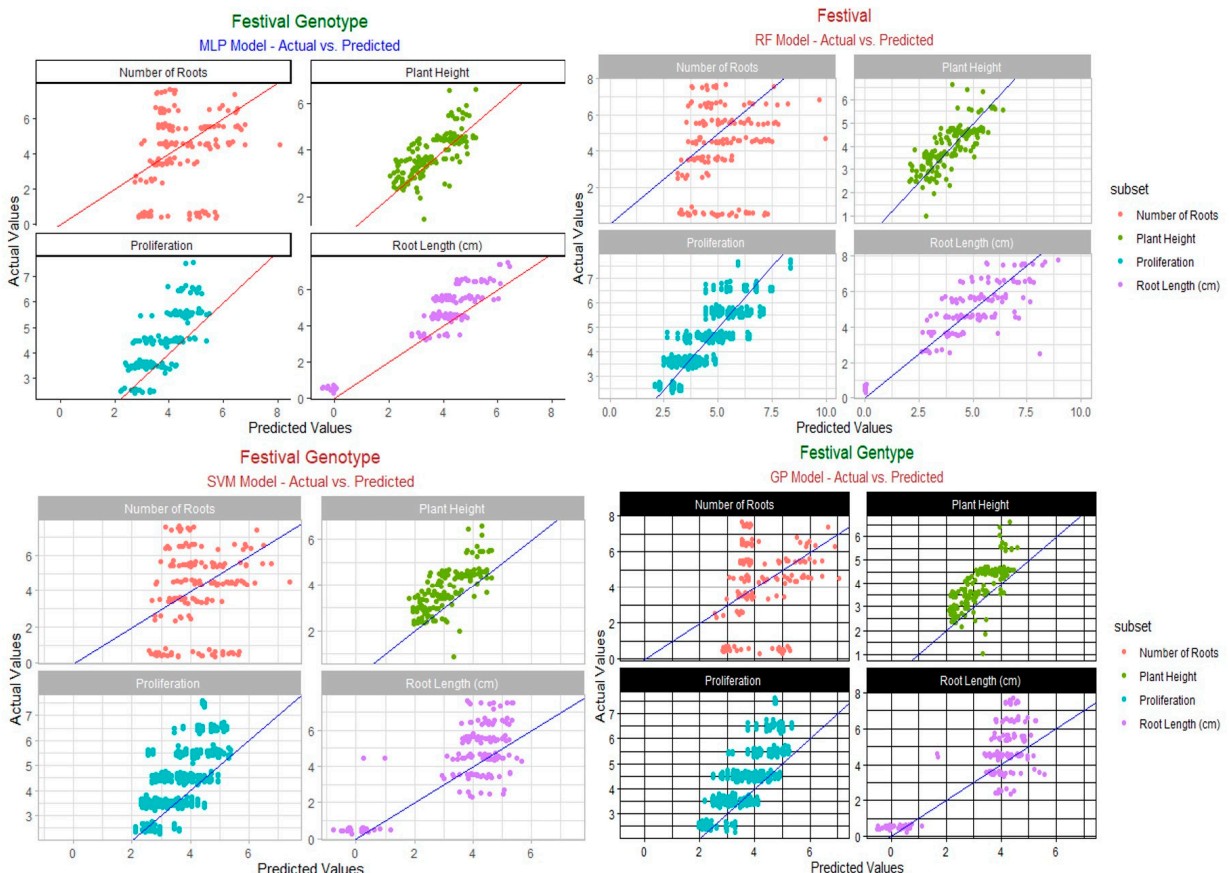

**Figure 1.** Predicted vs. actual value of "Rubygem" cultivar.

Table 7 also presents the models' *RMSE*, $R^2$, and *MAE* values for plant height, proliferation, number of roots, and root length for the "Festival" cultivar. The $R^2$ values for plant height scored 0.88 for RF, 0.86 for SVM, and 0.84 for RF, while GP showed the lowest $R^2$ at 0.78. The *RMSE* and *MAE* values varied by model and exhibited the order GP (0.10) = SVM (0.10) < RF (0.11) < MLP (0.12) for plant height. *MAE* also showed an order of GP = SVM < RF < MLP, with GP having the lowest *MAE*. In terms of the *RMSE*, $R^2$, and *MAE* values for proliferation, all tested and recorded models performed in the order MLP (0.11; 0.87; 0.10) > RF (0.14; 0.82; 0.12) > GP (0.15; 0.76; 0.12) > SVM (0.16; 0.67; 0.14). It can be deduced from these results that the MLP model was the best among the tested models in terms of its prediction of proliferation owing to its highest $R^2$ (0.87). In terms of the *RMSE*, $R^2$, and *MAE* values for the number of roots, all tested and recorded models performed in the order RF (0.11; 0.86; 0.09) > MLP (0.25; 0.63; 0.23)> SVM (0.23; 0.54; 0.21) > GP (0.28; 0.43; 0.25). For the number of roots in the "Festival" genotype, RF showed the highest $R^2$. The same thing happened with the *RMSE*, $R^2$, and *MAE* values for root length; all tested and recorded models performed in the order RF (0.10; 0.92; 0.63) > MLP (0.24; 0.52; 0.16) > GP (0.20; 0.47; 0.18) > SVM (0.22; 0.47; 0.19). The RF model outperformed the other models in terms of predicting root length. The overall performance of the models based on all variables (plant height, proliferation, number of roots, and root length) in the "Festival" cultivar can be ranked in the order RF > MLP > GP > SVM. The actual and predicted values of the "Festival" cultivar are shown in Figure 2.

**Table 7.** Validation of machine learning model algorithms for "Festival" cultivar.

|  |  | Plant Height | Proliferation | Number of Roots | Root Length |
|---|---|---|---|---|---|
| **MLP** | *RMSE* | 0.12 | 0.11 | 0.25 | 0.24 |
|  | $R^2$ | 0.84 | 0.87 | 0.63 | 0.52 |
|  | *MAE* | 0.11 | 0.10 | 0.23 | 0.16 |
| **SVM** | *RMSE* | 0.10 | 0.16 | 0.23 | 0.22 |
|  | $R^2$ | 0.86 | 0.67 | 0.54 | 0.47 |
|  | *MAE* | 0.08 | 0.14 | 0.21 | 0.19 |
| **RF** | *RMSE* | 0.11 | 0.14 | 0.11 | 0.10 |
|  | $R^2$ | 0.88 | 0.82 | 0.86 | 0.92 |
|  | *MAE* | 0.09 | 0.12 | 0.09 | 0.08 |
| **GP** | *RMSE* | 0.10 | 0.15 | 0.28 | 0.20 |
|  | $R^2$ | 0.78 | 0.76 | 0.43 | 0.47 |
|  | *MAE* | 0.0.08 | 0.12 | 0.25 | 0.18 |

MLP: multilayer perceptron; SVM: support vector machine; RF: random forest; GP: Gaussian process; $R^2$: coefficient of determination; *MAE*: mean absolute error; *RMSE*: root mean square error.

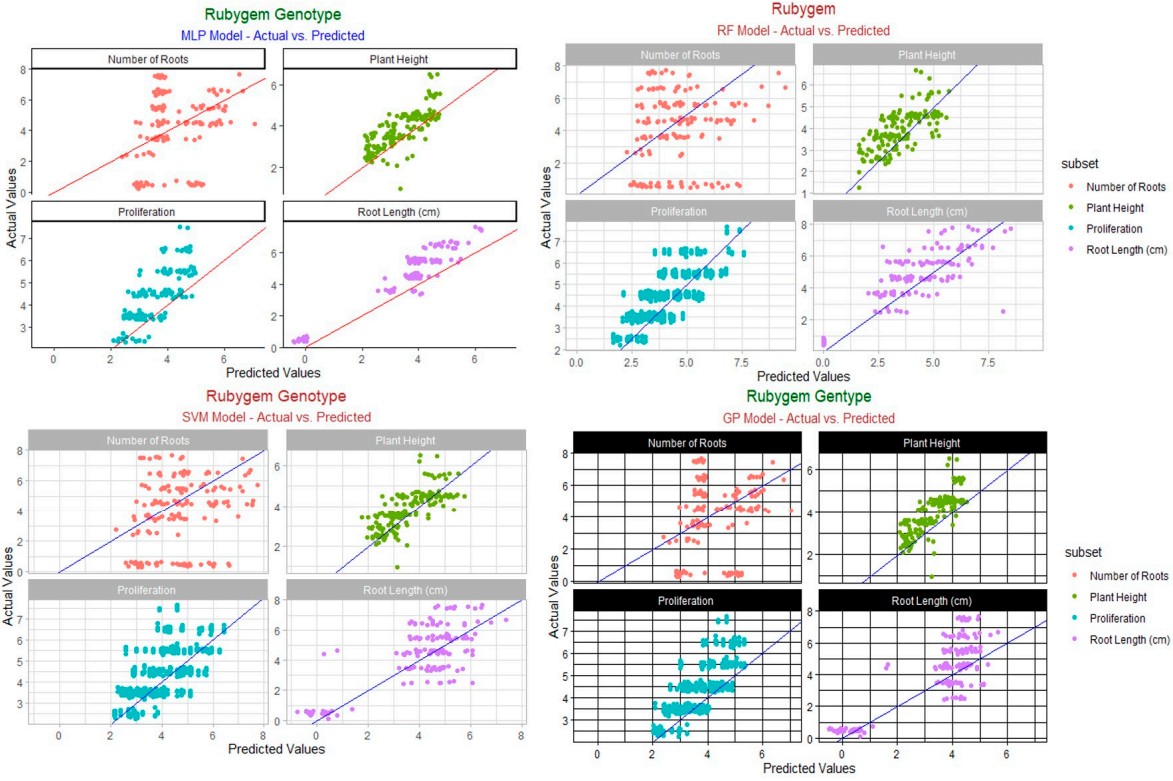

**Figure 2.** Predicted vs. actual value of "Festival" cultivar.

For the "Fortuna" cultivar, in terms of predicting plant height, the random forest (RF) model proved to be the most effective, with the lowest root mean square error (*RMSE*) of 0.12 and the highest $R^2$ value of 0.86. Conversely, the multilayer perceptron (MLP) model demonstrated a superior predictive ability for the proliferation metric, achieving the highest $R^2$ value of 0.80. Notably, the random forest model also exhibited the highest $R^2$ value, at 0.84, for predicting the number of roots. Additionally, for root length, the RF model outperformed the others, with the highest $R^2$ value of 0.89 and the lowest *RMSE* (0.11s) and mean absolute error (*MAE*) of 0.09. Overall, the random forest model consistently outperformed the other models across multiple metrics, highlighting its ability to capture complex relationships within genotypic data. While the MLP model demonstrated strength

in predicting proliferation, the support vector machine (SVM) and Gaussian process (GP) models showed competitive but slightly inferior performance across the evaluated variables (see Table 8). The actual and predicted values of the "Fortuna" cultivar are shown in Figure 3.

**Table 8.** Validation of machine learning model algorithms for "Fortuna" cultivar.

|  |  | Plant Height | Proliferation | Number of Roots | Root Length |
|---|---|---|---|---|---|
| **MLP** | *RMSE* | 0.12 | 0.16 | 0.27 | 0.22 |
|  | $R^2$ | 0.85 | 0.80 | 0.59 | 0.49 |
|  | *MAE* | 0.10 | 0.14 | 0.23 | 0.19 |
| **SVM** | *RMSE* | 0.11 | 0.16 | 0.26 | 0.22 |
|  | $R^2$ | 0.80 | 0.66 | 0.50 | 0.47 |
|  | *MAE* | 0.09 | 0.14 | 0.23 | 0.19 |
| **RF** | *RMSE* | 0.12 | 0.16 | 0.13 | 0.11 |
|  | $R^2$ | 0.86 | 0.79 | 0.84 | 0.89 |
|  | *MAE* | 0.11 | 0.14 | 0.12 | 0.09 |
| **GP** | *RMSE* | 0.12 | 0.16 | 0.27 | 0.21 |
|  | $R^2$ | 0.74 | 0.74 | 0.44 | 0.45 |
|  | *MAE* | 0.10 | 0.14 | 0.24 | 0.18 |

MLP: multilayer perceptron; SVM: support vector machine; RF: random forest; GP: Gaussian process; $R^2$: coefficient of determination; *MAE*: mean absolute error; *RMSE*: root mean square error.

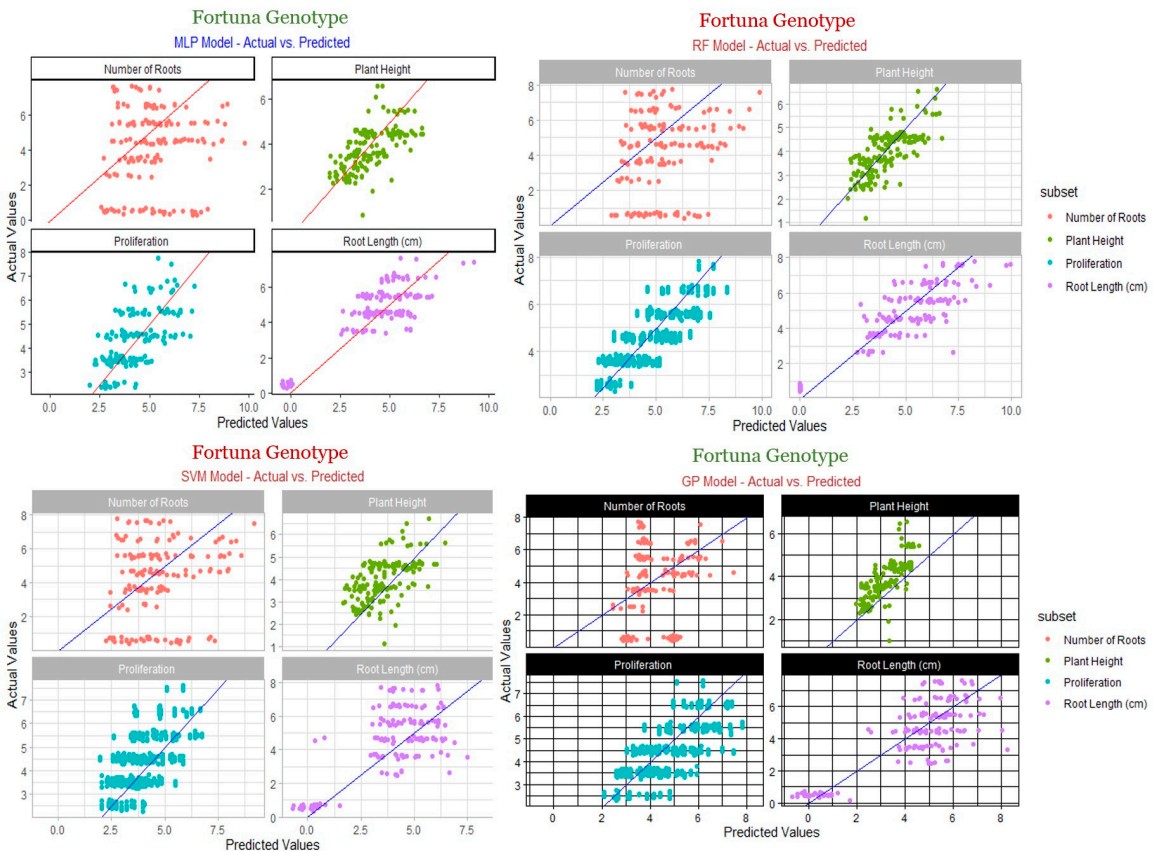

**Figure 3.** Predicted vs. actual value of "Fortuna" cultivar.

## 4. Discussion

Our experimental results reveal significant insights into the responses of three strawberry genotypes ("Festival", "Fortuna", and "Rubygem") to varying concentrations of

polyethylene glycol (PEG) simulating drought stress. In comparing our findings with studies by Hussein et al. [49], Mozafari et al. [50], Şener et al. [51], Josefi et al. [52], and Zahedi et al. [53], we can draw valuable comparisons to elucidate the implications of our results. The results indicate that PEG concentrations influence plant height, with the "Fortuna" cultivar reaching its maximum height in a PEG-free medium, while "Festival" showed the shortest height at 6 mg/L PEG. This is consistent with a study by Hussein et al. [49], where drought stress significantly reduced shoot length. Furthermore, the multiplication coefficients were highest in PEG-free conditions for "Fortuna" and "Festival," aligning with the observed robustness of these cultivars under normal conditions. Hussein et al. [49] investigated the impact of drought and salt stress on *F. ananasa* cultivars "Fortuna" and "Festival" by culturing apical meristems on MS medium without hormones, followed by exposure to 1.0 mg/L benzyl adenine with polyethylene glycol (PEG) concentrations of 0.0, 0.5, 1.0, 1.5, and 2%, and sodium chloride (NaCl) levels of 0.0, 500, 750, and 1000 mg/L. Drought stress significantly reduced chlorophyll content to 0.57 mg/g, shoot length to 1.50 cm, root length to 1.60 cm, and leaves per explant to 1.33. Salt stress at 1000 mg/L NaCl decreased chlorophyll content to 10.13 mg/g, shoot length to 2.10 cm, shoot dry biomass to 1.22 mg, root length to 2.13 cm, and root dry biomass to 0.96 mg. Stress conditions elevated proline to 3.95 μg/g, catalase to 4.75 μg/g, and peroxidase levels remained at 20.42 μg/g. These findings underscore the potential for in vitro selection of drought and salinity-tolerant strawberry plants. This mirrors the findings of Zahedi et al. [53], highlighting genotypic variations in drought tolerance. Zahedi et al. [53] examined the response of the "Camarosa" and "Gaviota" strawberry cultivars to varying levels of drought stress simulated by irrigation at 100, 75, 50, and 25% field capacity; significant differences in physiological and biochemical responses were observed. Drought stress decreased total chlorophyll, carotenoid, relative water, and phenolic content across both cultivars. However, "Gaviota" exhibited higher proline and hydrogen peroxide levels, indicating greater oxidative stress, whereas "Camarosa" demonstrated superior drought tolerance through increased soluble carbohydrates and antioxidant activities, highlighting the potential for genotypic selection in breeding drought-resistant strawberries.

Our study demonstrates unique patterns in root numbers, with "Rubygem" exhibiting the highest count in 6 mg/L PEG. The increase in root count at the maximum PEG concentration for all cultivars is reminiscent of the findings of Zahedi et al. [53], in whose study the accumulation of osmolytes and enhanced antioxidant enzyme activities were observed under severe drought stress. The decrease in root length at 6 mg/L PEG corresponds to Şener et al. [51], in whose study nano-silicon dioxide application mitigated reduced vegetative growth under drought stress. Our data on root rates at different PEG concentrations align with a study by Josefi et al. [52], emphasizing the influence of jasmonic acid (JA) on morphophysiological and biochemical characteristics under PEG-induced water stress. The higher root rate for "Festival" at 4 mg/L PEG indicates its enhanced response to JA application, while the lowest rate for "Fortuna" at 2 mg/L PEG suggests varied responses among cultivars.

Hussein et al. [49] and Mozafari et al. [50] both emphasize the negative impacts of drought stress on various parameters, highlighting the importance of mitigating strategies. The application of iron nano-particles and salicylic acid, as discussed by Mozafari et al. [50], resonates with our findings on the influence of PEG on strawberry growth traits. Additionally, Şener et al. [51] and Josefi et al. [52] provide evidence of the positive effects of nano-silicon dioxide and jasmonic acid in enhancing drought stress tolerance. Zahedi et al. [53] compare two strawberry cultivars under different drought conditions, supporting our emphasis on genotypic variations. The "Camarosa" cultivar's higher tolerance, reflected in its physiological responses, suggests potential breeding strategies to develop drought-tolerant strawberries. Our results contribute to the growing knowledge of strawberry responses to drought stress. The variations observed among genotypes and the potential mitigating effects of external factors underscore the importance of tailored approaches in strawberry cultivation under stress conditions. Future research should

continue exploring the underlying mechanisms and developing practical applications to enhance strawberry resilience in changing environmental conditions.

Our study employed multilayer perceptron (MLP), support vector machine (SVM), Gaussian process (GP), and random forest (RF) models. A similar variety of models has been used in the literature, including support vector regression (SVR), XGBoost, and MLP.

The selection between MLP, SVM, GP, and RF hinges on the specific data characteristics and application requirements. MLP excels in approximating complex functions across diverse applications, but it may overfit and needs careful hyperparameter tuning. SVM performs well in high-dimensional spaces and with nonlinear data, yet it is sensitive to kernel and regularization choices. GP offers flexibility and precise uncertainty estimates for smaller datasets but struggles with scalability. RF is robust against overfitting and suits large, high-dimensional datasets, though it may lag in real-time prediction and presents interpretative challenges. The optimal model choice necessitates a balanced consideration of these strengths and limitations, tailored to the task at hand [54].

RF consistently demonstrated superior performance in multiple studies, aligning with our findings. For instance, Kirtis et al. [39] highlighted RF's high performance in predicting in vitro plant characteristics, especially in shoot count and length. The evaluation metrics in our study included root mean square error (*RMSE*), coefficient of determination ($R^2$), and mean absolute error (*MAE*). These metrics have also been used in the literature, with $R^2$ being a crucial indicator of predictive accuracy. The literature consistently emphasizes the importance of lower *RMSE* and *MAE* values, similar to our interpretation.

In the study by Kirtis et al. [39], the RF model excelled in predicting shoot count and length for desi chickpeas, while XGBoost outperformed in shoot count. Similarly, our study found RF to be the best model across all variables (plant height, proliferation, number of roots, and root length). This demonstrates a consistent RF efficacy pattern across different plant species and characteristics. Aasim et al. [38] and Rezaei et al. [41] applied ML models to optimize plant tissue culture protocols. Aasim et al. [54] utilized MLP to predict shoot regeneration in common beans, while Rezaei et al. [55] employed a genetic algorithm (GA) in conjunction with ML models to optimize phytohormone concentrations in petunia callogenesis. Our study, while not explicitly focusing on optimization, corroborates the potential of ML in achieving efficient tissue culture protocols. Sadat-Hosseini et al. [56] compared three ML approaches—multilayer perceptron neural network (MLPNN), k-nearest neighbors (KNN), and gene expression programming (GEP)—in predicting the in vitro proliferation of Persian walnuts. Our study did not explore these specific techniques, but the comparison underscores the diversity of ML methods available for plant-related predictions.

Despite the promising results, it is crucial to acknowledge challenges such as model interpretability, overfitting, and the need for large datasets. Future research could focus on addressing these challenges and exploring emerging ML techniques. Our study aligns with the existing literature, emphasizing the efficacy of RF models in predicting various plant characteristics. The comparison provides a comprehensive overview of ML applications in plant tissue culture, offering insights for future research and potential optimizations in diverse plant species and traits.

## 5. Conclusions

The present study provides valuable insights into the responses of three strawberry genotypes to varying PEG concentrations simulating drought stress. PEG significantly influenced plant height, multiplication coefficients, root numbers, and root rates, indicating genotype-specific reactions. Comparison with the existing literature supported our findings, highlighting the importance of mitigating strategies for drought stress. The application of machine learning models, especially random forest, which demonstrated consistent efficacy in predicting plant characteristics with an accuracy of 91.164%, was pivotal in accurately predicting the impact of drought stress on strawberry plants. Our analysis reveals significant variations in the performance of machine learning (ML) models across different strawberry cultivars, with the random forest (RF) model consistently show-

casing superior accuracy in predicting key phenotypic traits such as root length and the number of roots, especially in the "Festival" and "Fortuna" cultivars. This demonstrates the model's robustness and adaptability to various genetic backgrounds. For the "Rubygem" cultivar, the study found the multilayer perceptron (MLP) model to be highly effective in predicting proliferation, while the Gaussian process (GP) model was particularly accurate in estimating plant height. However, challenges such as model interpretability and the need for larger datasets should be addressed in future research. These results underline the considerable potential of employing ML models, notably the RF model, for precise phenotypic predictions in strawberries. Such advancements hold promise for the agricultural sector, potentially transforming breeding and cultivation strategies through enhanced efficiency and accuracy. This study not only delineates the comparative strengths and limitations of each model within the realm of agricultural forecasting but also paves the way for future research aimed at refining these predictive models for greater precision and utility in agricultural applications. This study contributes to the growing knowledge of plant responses to stress conditions and suggests potential breeding strategies for developing drought-tolerant strawberries. Future research should focus on elucidating underlying mechanisms and practical applications to enhance strawberry resilience in changing environmental conditions.

**Funding:** This research was funded by Erciyes University Scientific Projects Units, grant number FBA-2020-9756.

**Institutional Review Board Statement:** Not applicable

**Data Availability Statement:** The data presented in this study are available upon reasonable request from the corresponding author.

**Acknowledgments:** The authors would like to thank Eugene Steele, professional English editor of Erciyes University, for editing the manuscript in English. We thank the Office of the Dean for Research at Erciyes University for providing the necessary infrastructure and laboratory facilities at the ArGePark Research Building.

**Conflicts of Interest:** The author declares no conflicts of interest.

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
