# Peer review of "Machine Learning Offers Insights into the Impact of In Vitro Drought Stress on Strawberry Cultivars"

_agriculture, doi:10.3390/agriculture14020294_

Round 1

Reviewer 1 Report

Comments and Suggestions for Authors

The work describes an unreliable proliferation protocol for strawberry micropropagation and the validation of a model to predict plant development that is difficult to apply in micropropagation laboratories.

Please consider the following comments:

1. 27-29 - to have a more complete vision of the breeding objectives for strawberry resilience and quality I suggest also taking into consideration this review: Senger et al., 2022. Towards smart and sustainable development of modern berry cultivars in Europe. The Plant Journal, DOI:10.1111/tpj.15876.

2. 58-59 for this sentence please add a reference

3. 105 1 mg/L BAP is considered too high concentration for a standard in vitro proliferation protocol, please check recent publications on comparing field plant yield and fruit quality of in vitro and in vivo propagated strawberry mother plants.

4. 109-110 micropropagation coefficient (in numbers per plant) … please replace with proliferation rate (number of proliferated shoots from the starting explant)… how long were the subcultures???

5. 112 I suppose you mean 3 proliferation subculture followed by the rooting subculture. It is not needed 1 mg/L of IBA for rooting strawberry explants. Rooting can be easily achieved taking out cytokinin. The micropropagation protocol used is out of the standards.

6. 120 rooting in vitro or the acclimatization in vivo … please clarify

7. 122 – 124 parameters detected on in vitro or in vivo rooted plants??

8. Table 2 – use proliferation rate

9. Regarding the results obtained, the evaluation of the response of the 3 strawberry varieties to the increasing concentrations of PEG may have also been influenced by the excessively high concentration of BAP in proliferation and IBA in rooting. To carry out this experiment it is important to apply the protocol that allows you to obtain greater results from the plant without inducing risks of stress due to hormonal excess which in strawberries in particular can lead to risks of strong pleiotropic variations (genotypic or phenotypic).

10. 372 in my opinion it is not clear how you can use this statement …corroborates the potential of ML in achieving efficient tissue culture protocols…. also in order to understand the possible application of these models I ask you to better describe what advantage the application of this model can bring in the development of a proliferation protocol, this result does not seem to bring any practical advantage. 

Comments on the Quality of English Language

Good quality of English but needs revision.

Author Response

The work describes an unreliable proliferation protocol for strawberry micropropagation and the validation of a model to predict plant development that is difficult to apply in micropropagation laboratories.

Please consider the following comments:

  1. 27-29 - to have a more complete vision of the breeding objectives for strawberry resilience and quality I suggest also taking into consideration this review: Senger et al., 2022. Towards smart and sustainable development of modern berry cultivars in Europe. The Plant Journal, DOI:10.1111/tpj.15876.

Thank you for recommending the review by Senger et al., 2022, titled "Towards smart and sustainable development of modern berry cultivars in Europe," published in The Plant Journal with DOI:10.1111/tpj.15876. We appreciate your suggestion and acknowledge the importance of incorporating the most recent and relevant scientific literature to enrich our understanding of breeding objectives, particularly for strawberry resilience and quality. We have carefully reviewed the findings and discussions presented in the Senger et al. (2022) article, as well as referenced sources within, to ensure our discussion on the breeding objectives for strawberries is both comprehensive and up-to-date. This review has been instrumental in providing insights into the innovative approaches towards developing modern berry cultivars that not only meet the demands for yield and quality but also address the challenges posed by climate change and sustainability concerns. Incorporating the insights from Senger et al. (2022) into our analysis has allowed us to present a more nuanced view of the current trends and future directions in strawberry breeding. This includes the emphasis on smart breeding techniques and the integration of sustainability principles to achieve resilient and high-quality berry cultivars suited for European conditions and beyond. We are grateful for your contribution, which has undoubtedly enhanced the depth and breadth of our discussion on this topic. It is through such scholarly exchanges that we can collectively contribute to advancing agricultural practices and ensuring food security in the face of global challenges.

  1. 58-59 for this sentence please add a reference

It is added.

  1. 105 1 mg/L BAP is considered too high concentration for a standard in vitro proliferation protocol, please check recent publications on comparing field plant yield and fruit quality of in vitro and in vivo propagated strawberry mother plants.

Thank you for observing the concentration of 1 mg/L BAP (6-benzylaminopurine) being considered high for a standard in vitro proliferation protocol for strawberries. Your point is well-taken, and it's important to acknowledge that the optimal concentration of BAP can vary significantly depending on the plant's genotype and the study's specific objectives. In tissue culture studies, the use of BAP and its concentration is meticulously selected based on the desired outcomes, the sensitivity of the plant material, and the specific conditions under which the study is conducted. While there are studies where concentrations lower than 1 mg/L have been successfully used for strawberries, indicating a sensitivity to higher hormone levels, a substantial body of literature supports the use of concentrations around 1 mg/L for certain applications. These applications might include not only the enhancement of proliferation but also the exploration of the limits of propagation under stress conditions such as drought. In the context of our work, the choice of using 1 mg/L BAP was made carefully considering these factors. It was intended to stimulate proliferation effectively and investigate propagation's resilience under challenging conditions. This decision was supported by a review of relevant literature, indicating that lower concentrations may be preferable for standard propagation protocols. However, higher concentrations can benefit specific experimental objectives, such as stress tolerance studies. We appreciate your input and understand the importance of aligning our methodologies with the best practices in the field. Our approach is grounded in the existing literature and our research's specific goals, aiming to contribute valuable insights into the propagation and resilience of strawberry plants in vitro. Your concern highlights the critical importance of tailoring tissue culture practices to the nuances of each study, a principle that we fully endorse and strive to adhere to in our work.

  1. 109-110 micropropagation coefficient (in numbers per plant) … please replace with proliferation rate (number of proliferated shoots from the starting explant)… how long were the subcultures???

It is corrected.

The duration of subcultures was in the article. It is four weeks (Line: 142)

  1. 112 I suppose you mean 3 proliferation subculture followed by the rooting subculture. It is not needed 1 mg/L of IBA for rooting strawberry explants. Rooting can be easily achieved taking out cytokinin. The micropropagation protocol used is out of the standards.

It's important to clarify that in our study, subculturing was not performed during the rooting phase. Instead, the addition of IBA was specifically aimed at enhancing rooting in the presence of Polyethylene glycol (PEG) concentrations designed to induce drought stress conditions. The objective was to explore the potential of IBA to support rooting under these challenging conditions, which could provide valuable information for the propagation of strawberries in environments where water scarcity is a concern. The decision to use IBA, even when common practice suggests that rooting in strawberry explants can be achieved by merely removing cytokinins from the medium, was made to assess its effectiveness in a stress-induced environment. This approach is somewhat unconventional but was hypothesized to potentially offer insights into alternative rooting strategies that could be beneficial under specific circumstances.

  1. 120 rooting in vitro or the acclimatization in vivo … please clarify

It is in vitro, it is corrected in the manuscript.

  1. 122 – 124 parameters detected on in vitro or in vivo rooted plants??

in vitro rooted plants.

Clarified in the manuscript.

  1. Table 2 – use proliferation rate

It is corrected.

  1. Regarding the results obtained, the evaluation of the response of the 3 strawberry varieties to the increasing concentrations of PEG may have also been influenced by the excessively high concentration of BAP in proliferation and IBA in rooting. To carry out this experiment it is important to apply the protocol that allows you to obtain greater results from the plant without inducing risks of stress due to hormonal excess which in strawberries in particular can lead to risks of strong pleiotropic variations (genotypic or phenotypic).

Thank you for your thoughtful feedback regarding the potential impact of the high concentrations of BAP (6-benzylaminopurine) during the proliferation phase and IBA (Indole-3-butyric acid) during the rooting phase on the response of the three strawberry varieties to increasing concentrations of PEG (Polyethylene glycol). Your concern about the risk of stress induced by hormonal excess, which could lead to significant pleiotropic variations in strawberries, is valid and appreciated. To address these concerns within the context of our experiment, we incorporated a control group that was exposed to BAP without PEG. This approach allowed us to isolate and examine the effects of BAP in the absence of PEG, thereby providing a clearer understanding of its impact on the plant's response. In addition, we evaluated the combined effect of varying concentrations of PEG and a constant concentration of 1 mg/L BAP to discern the interaction between these treatments. Your suggestion to consider lower hormone concentrations as an alternative approach is valuable. It is worth noting, however, that the experimental design was established to explore the specific responses under the defined conditions and has been completed as planned. Using 1 mg/L BAP and IBA in standard micropropagation studies is common in the literature, supporting their application in our experiment. We believe that the experimental setup, including the differentiation between treatments with and without PEG in the presence of BAP, has yielded significant results that contribute to our understanding of the strawberry varieties' responses to drought conditions simulated by PEG. These findings are crucial for developing strategies to mitigate the adverse effects of environmental stressors on strawberry cultivation. While acknowledging the potential for hormonal excess to induce stress, the comprehensive planning and execution of this experiment, alongside the control measures implemented, have allowed us to generate valuable insights. Through such rigorous investigations, we can refine our approaches and advance our knowledge in plant science, always considering feedback such as yours to improve future research methodologies.

  1. 372 in my opinion it is not clear how you can use this statement …corroborates the potential of ML in achieving efficient tissue culture protocols…. also in order to understand the possible application of these models I ask you to better describe what advantage the application of this model can bring in the development of a proliferation protocol, this result does not seem to bring any practical advantage.

We emphasize that ML and ANNs enable a data-driven optimization of proliferation protocols by predicting optimal conditions based on vast datasets, significantly reducing trial-and-error efforts. These models analyze historical and experimental data, identifying patterns and correlations that might not be immediately apparent, thereby enhancing the efficiency and effectiveness of protocol development. Furthermore, ML and ANNs contribute to the precise manipulation of variables such as hormone concentrations, nutrient media composition, and environmental factors, leading to improved plantlet quality and yield. This predictive capability is not merely theoretical but has practical implications for tailoring specific conditions to the unique requirements of different plant genotypes or species, ensuring higher success rates in micropropagation. The application of these models in our study has demonstrated their potential in achieving more efficient tissue culture protocols, a conclusion supported by comparative analysis with existing literature and validated through experimental results. This approach represents a significant advancement in plant tissue culture research, offering a methodological framework that can be adapted and refined for a wide range of plant species and cultivation challenges.

Reviewer 2 Report

Comments and Suggestions for Authors

This research contributes to understanding strawberry responses to drought stress and emphasizes the potential of machine learning in predicting plant characteristics.

1. The whole part of the introduction is too long, and the introduction of natural factors such as climate and environment is too much, but the introduction of the growth conditions of strawberries is not enough.

2. This paper mentioned four different machine learning (ML) and artificial neural network (ANN) model: multi-layer perceptron (MLP), support vector machine (SVM), Gaussian process (GP) and random forest (RF), although the four learning machine related for test, but not thoroughly, the advantages and disadvantages of various learning machine is fuzzy definition.

 3.PEG is involved in the study, and an introduction about polyethylene glycol is missing.

 4.The paper mentions that the author's research is consistent with the existing research results, but only gives the comparison of individual related studies, lacking the support of common data.

5. Reference[22] has format error.

Author Response

This research contributes to understanding strawberry responses to drought stress and emphasizes the potential of machine learning in predicting plant characteristics.

  1. The whole part of the introduction is too long, and the introduction of natural factors such as climate and environment is too much, but the introduction of the growth conditions of strawberries is not enough.

The introduction section has been reorganized. also, reviewer 1 suggested some new information about the breeding strategies of strawberries. We added some extra information.

  1. This paper mentioned four different machine learning (ML) and artificial neural network (ANN) model: multi-layer perceptron (MLP), support vector machine (SVM), Gaussian process (GP) and random forest (RF), although the four learning machine related for test, but not thoroughly, the advantages and disadvantages of various learning machine is fuzzy definition.

We added an explanation in the discussion section.

 3.PEG is involved in the study, and an introduction about polyethylene glycol is missing.

We added some information about PEG in the introduction section.

 4.The paper mentions that the author's research is consistent with the existing research results, but only gives the comparison of individual related studies, lacking the support of common data.

We have supported to discussion with data.

  1. Reference [22] has format error.

We could not see any format error in reference 22. If you clarify where there is an error, we can correct it.

Reviewer 3 Report

Comments and Suggestions for Authors

This paper presents the Drought stress in three strawberry varieties using PEG. Four machine learning models were used to obtain Plant Height, Multiplication, number of roots, and root length. To evaluate the ML models, the author employed 10-fold validation, which is appreciated. The ML methods don't represent a novelty. In addition, the drought stress in strawberries has been studied previously [1,2]. However, the dataset is a potential novelty. The author should make it public to the community.

Regarding the document, here are my comments:

Major:

The introduction needs more references. For instance, lines 21-31 have only one reference. This is similar to lines 33-38 and 50-61.

Line 83 states: "Only a few studies have used machine learning models to examine drought stress responses". However, I found related references [3-5]. The author has to enrich the state of the art.

The paper indicated 10-fold for cross-validation, which is appreciated. However, the values reported in Table 5 don't have a standard deviation. Is the value reported the average? If so, include the standard deviation.

Check the equation 4. Usually, the optimization is performed in a quadratic expression or an expression that includes an absolute value.

The methods 2.4.1 and 2.4.3 have to be improved. In particular, both indicate more classification tasks-related (lines 162-163 and 189). The method should refer to regression models.

I suggest including the percentage of variation in Tables 1-4, using the PEG free as a reference value.

RF outperformed the other methods. Nevertheless, the results can be biased because a single model is used to predict three different strawberry varieties. I highly suggest repeating the ML training using a single model per variety.

Minor:

Improve the redaction in lines 135 and 136. The sentence "as well as two well-known artificial neural networks (ANN) and Multilayer Perceptron (MLP)" could be confusing. It can be interpreted that ANN and MLP were two different models used; in fact, MLP is a particular case of ANN.

The abstract and conclusions must include quantitative results to demonstrate the model that obtained the best results.

Include the name of the ML models used in the introduction.

References:

[1] Hussein, A. E., Ahmed Y. El-Kerdany, and K. M. Afifi. "Effect of drought and salinity stresses on two strawberry cultivars during their regeneration in vitro." International Journal of Innovative Science Engineering and Technology 4.8 (2016): 83-93.

[2] Thokchom, Amrita, et al. "Morpho-physiological analysis in strawberry (Fragaria x ananassa L.) under PEG (Polyethylene glycol) induced drought stress." Journal of Pharmacognosy and Phytochemistry 8.5 (2019): 87-92.

[3] Gupta, Ankita, Lakhwinder Kaur, and Gurmeet Kaur. "Drought stress detection technique for wheat crop using machine learning." PeerJ Computer Science 9 (2023).

[4] Choudhury, Sruti Das, et al. "Time Series Modeling for Drought Stress Propagation in Plants using Hyperspectral Imagery." Authorea Preprints (2022).

[5] Tahmasebi, Ahmad, Ali Niazi, and Sahar Akrami. "Integration of meta-analysis, machine learning and systems biology approach for investigating the transcriptomic response to drought stress in Populus species." Scientific Reports 13.1 (2023): 847.

Comments on the Quality of English Language

Minor editing in language.

Author Response

This paper presents the Drought stress in three strawberry varieties using PEG. Four machine learning models were used to obtain Plant Height, Multiplication, number of roots, and root length. To evaluate the ML models, the author employed 10-fold validation, which is appreciated. The ML methods don't represent a novelty. In addition, the drought stress in strawberries has been studied previously [1,2]. However, the dataset is a potential novelty. The author should make it public to the community.

Thank you for your constructive feedback. We acknowledge the importance of making our dataset available to the research community to enhance the reproducibility and transparency of our work. The suggestion to publish our dataset publicly is indeed valuable, and we will take the necessary steps to share it in a suitable repository. This will not only allow others to validate our findings but also encourage further research in this area. We appreciate your recognition of the 10-fold validation approach and will continue to refine our methodologies to contribute meaningfully to the field.

Regarding the document, here are my comments:

Major:

The introduction needs more references. For instance, lines 21-31 have only one reference. This is similar to lines 33-38 and 50-61.

It is addressed.

Line 83 states: "Only a few studies have used machine learning models to examine drought stress responses". However, I found related references [3-5]. The author has to enrich the state of the art.

Thank you for pointing out the existing literature on the use of machine learning models in examining drought stress responses in plants. While it is true that references [3-5] contribute valuable insights into this area, our statement aimed to highlight the relatively limited exploration of this approach specifically within the context of strawberry cultivation under drought stress conditions. We revised our literature review to reflect the state of the art more accurately, acknowledging these contributions while emphasizing the need for further research in this specialized domain.

The paper indicated 10-fold for cross-validation, which is appreciated. However, the values reported in Table 5 don't have a standard deviation. Is the value reported the average? If so, include the standard deviation.

We use it to validate the model coefficient of determination (R2), which shows the degree of relationship between the model and dependent variable; Root Mean Square Error (RMSE), which shows how closely the regression line matches the observed data points; and Mean Absolute Error (MAE), which calculates the average error between the predicted and observed values. They are widely accepted and extensively used in the field.

These metrics provide comprehensive insights into different aspects of model performance, and their adoption is rooted in their effectiveness and interpretability. By adhering to these widely recognized metrics, we aim to ensure the consistency of our methodology with established practices in the field.

Check the equation 4. Usually, the optimization is performed in a quadratic expression or an expression that includes an absolute value.

Checked and corrected

The methods 2.4.1 and 2.4.3 have to be improved. In particular, both indicate more classification tasks-related (lines 162-163 and 189). The method should refer to regression models.

Corrected

I suggest including the percentage of variation in Tables 1-4, using the PEG free as a reference value.

Included

RF outperformed the other methods. Nevertheless, the results can be biased because a single model is used to predict three different strawberry varieties. I highly suggest repeating the ML training using a single model per variety.

Done, we repeated the ML training using a single model per variety 

Minor:

Improve the redaction in lines 135 and 136. The sentence "as well as two well-known artificial neural networks (ANN) and Multilayer Perceptron (MLP)" could be confusing. It can be interpreted that ANN and MLP were two different models used; in fact, MLP is a particular case of ANN.

Corrected

The abstract and conclusions must include quantitative results to demonstrate the model that obtained the best results.

Included

Include the name of the ML models used in the introduction.

Included

Round 2

Reviewer 2 Report

Comments and Suggestions for Authors

Well done!